# Open-set Label Noise Can Improve Robustness Against Inherent Label Noise

**Hongxin Wei**[1]    **Lue Tao**[2,3]    **Renchunzi Xie**[1*]    **Bo An**[1]

[1]School of Computer Science and Engineering, Nanyang Technological University, Singapore
[2] College of Computer Science and Technology,
Nanjing University of Aeronautics and Astronautics, China
[3] MIIT key Laboratory of Pattern Analysis and Machine Intelligence, China
{hongxin001,xier0002}@e.ntu.edu.sg    tlmichael@nuaa.edu.cn    boan@ntu.edu.sg

## Abstract

Learning with noisy labels is a practically challenging problem in weakly super-vised learning. In the existing literature, open-set noises are always considered to be poisonous for generalization, similar to closed-set noises. In this paper, we empirically show that open-set noisy labels can be non-toxic and even benefit the robustness against inherent noisy labels. Inspired by the observations, we propose a simple yet effective regularization by introducing Open-set samples with Dynamic Noisy Labels (ODNL) into training. With ODNL, the extra capacity of the neural network can be largely consumed in a way that does not interfere with learning patterns from clean data. Through the lens of SGD noise, we show that the noises induced by our method are random-direction, conflict-free and biased, which may help the model converge to a flat minimum with superior stability and enforce the model to produce conservative predictions on Out-of-Distribution instances. Extensive experimental results on benchmark datasets with various types of noisy labels demonstrate that the proposed method not only enhances the performance of many existing robust algorithms but also achieves significant improvement on Out-of-Distribution detection tasks even in the label noise setting.

## 1   Introduction

The success of deep neural networks (DNNs) heavily relies on a large number of training instances with fully accurate labels [16]. In real-world applications, it is expensive and time-consuming to collect such large-scale datasets. To alleviate this problem, some surrogate methods are commonly used to improve labelling efficiency, such as online queries [3] and crowdsourcing [74]. These cheap but imperfect methods usually suffer from unavoidable noisy labels that can be easily memorized by Deep Neural Networks (DNNs), leading to poor generalization performance [2, 79]. Therefore, designing robust algorithms against noisy labels is of great practical importance.

Label noise can be separated into two categories: 1) closed-set noise, where instances with noisy labels have true class labels within the noisy label set [15, 24, 50, 51, 60, 63]. 2) open-set noise, where instances with noisy labels have some true class labels outside the noisy label set [30, 54, 61, 67]. In the existing literature, open-set noises are always considered to be harmful to the training of DNNs like closed-set noises [30, 61, 54, 67]. For example, ILON [61] reduces the effect of open-set noises by using noisy label detector and contrastive loss to pull away noisy samples from clean samples. Extended T [67] uses an extended transition matrix to mitigate the impacts of open-set noises and EvidentialMix [54] filters out the open-set examples by modelling the loss distribution. Therefore, there arises a question to be answered: *Do open-set noises always play a negative role in the training?*

---

[*]Corresponding author.

35th Conference on Neural Information Processing Systems (NeurIPS 2021).

In this work, we answer this question by experiments with an open-set auxiliary dataset. We empirically show that additional open-set noises can be harmless to generalization if the number of open-set noises is sufficiently large. More surprisingly, we observe that the open-set noises can even benefit the robustness of neural networks against noisy labels from the original training dataset, while the closed-set counterpart could not. Based on the "insufficient capacity" hypothesis [2], an intuitive explanation for this phenomenon is that increasing the number of open-set auxiliary samples slows down the fitting of inherent noises. In other words, fitting open-set label noises consumes extra capacity of the network, thereby reducing the memorization of inherent noisy labels. The details of experimental results are presented in Subsection 2.2.

Inspired by the above observations, we propose an effective and complementary method to improve robustness against noisy labels. The high-level idea is to introduce Open-set samples with Dynamic Noisy Labels (ODNL) into training. In each epoch, we assign random labels to instances of open-set auxiliary dataset by uniformly sampling from the label set. In other words, the label of each open-set sample is independently random and is constantly changing over training epochs. In this manner, ODNL consistently produce "benign hard examples" to consume the extra representation capacity of neural networks, thereby preventing the neural network from overfitting inherent noisy labels.

To further understand the effect of ODNL, we provide an analysis from the lens of SGD noise [7, 65], and formalize the differences and connections between our regularization and related methods. From the analysis, we show that the SGD noises induced by our method are 1) random-direction, helping the model converge to a flat minimum; 2) conflict-free, leading to good stability for the training; 3) biased, enforcing the model to produce more conservative predictions on OOD examples. In this manner, our method not only improves the robustness against noisy labels, but also provides benefits for OOD detection performance, which is also essential for the deployment of deep learning in safety-critical applications.

To the best of our knowledge, we are the first to explore the benefits of open-set auxiliary dataset in learning with noisy labels. To verify the efficacy of our regularization, we conduct extensive experiments on both simulated and real-world noisy datasets, including CIFAR-10, CIFAR-100 and Clothing1M datasets. In summary, the proposed method can be easily adapted for a wide range of existing algorithms to prevent overfitting noisy labels. Furthermore, experimental results validate that our regularization could also achieve impressive performance for detecting OOD examples, even if the labels of training dataset are noisy.

## 2    Problem setting and motivation

In this section, we first briefly introduce the problem setting of learning from training data with noisy labels. Then we start in Subsection 2.2 with an observational study highlighting the effect of open-set noisy examples from the auxiliary dataset on generalization and robustness against noisy labels.

### 2.1    Preliminaries: learning with inherent noisy labels

In this work, we consider multi-class classification with $k$ classes. Let $\mathcal{X} \subset \mathbb{R}^d$ be the feature space and $\mathcal{Y} = \{1, \ldots, k\}$ be the label space, we suppose the training dataset with $N$ samples is given as $\{\boldsymbol{x}_i, y_i\}_{i=1}^N$, where $\boldsymbol{x}_i \in \mathcal{X}$ is the $i$-th instance sampled from a certain data-generating distribution $\mathcal{D}_{\text{train}}$ in an i.i.d. manner and $y_i \in \{1, \ldots, k\}$ represents its observed label that might be different from the ground truth. To prevent confusion, the noisy labels in the training dataset are referred to as *inherent noisy labels*. A classifier is a function that maps the feature space to the label space $f : \mathcal{X} \to \mathbb{R}^k$ with trainable parameter $\boldsymbol{\theta} \in \mathbb{R}^p$. In this paper, we consider the common case where the function $f$ is a DNN with the softmax output layer. The objective function of standard training can be represented as a cross-entropy loss:

$$\mathcal{L}_{CCE} = \mathbb{E}_{\mathcal{D}_{\text{train}}}\left[\ell\left(f(\boldsymbol{x}; \boldsymbol{\theta}), y\right)\right] \simeq -\frac{1}{N}\sum_{i=1}^N \boldsymbol{e}^{y_i} \log f\left(\boldsymbol{x}_i; \boldsymbol{\theta}\right), \tag{1}$$

where $f\left(\boldsymbol{x}_i; \boldsymbol{\theta}\right)$ denotes the output of the classifier $f$ with $\boldsymbol{\theta}$ for $\boldsymbol{x}_i$ and $\boldsymbol{e}^{y_i} := (0, \ldots, 1, \ldots, 0)^\top \in \{0, 1\}^k$ is a one-hot vector and only the $y_i$-th entry of $\boldsymbol{e}^{y_i}$ is 1.

In addition to the training data, we consider there is an unlabelled auxiliary dataset consisting of open-set instances $\{\widetilde{\boldsymbol{x}}_i\}_{i=1}^M$. Specifically, open-set instances are sampled from $\mathcal{D}_{\text{out}}$, disjoint from

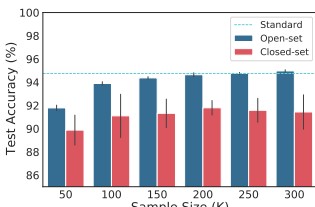
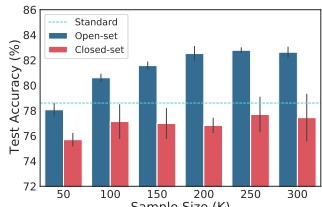
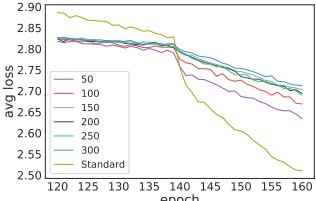

(a) Learning with Clean data     (b) Learning with inherent noises     (c) Losses of inherent noises

Figure 1: Average test accuracy (%) on CIFAR-10 with (a) clean labels (b) inherent noisy labels under different auxiliary dataset sizes. We use "open-set" and "closed-set" to denote two types of auxiliary datasets. The results of standard training are also reported for comparison. All experiments are repeated five times and the standard deviations are presented as error bars. (c) Training losses of inherent noises around the 140th epoch under different auxiliary dataset sizes (K).

$\mathcal{D}_{\text{train}}$. Therefore, they are also called Out-of-Distribution (OOD) instances. In real applications, it is easy to get such an auxiliary dataset, which is commonly used in OOD detection tasks [20, 48].

## 2.2 Motivation: the role of open-set label noise in generalization and robustness

To analyze the effects of open-set noisy labels, we build an open-set noisy dataset with the unlabelled auxiliary dataset. For each open-set instance, we assign a fixed random label $\widetilde{y}_i$ uniformly sampled from the label set $\{1, \ldots, k\}$. In each iteration, we concatenate the two batches that are sampled from the training dataset and the open-set noisy dataset respectively, then perform standard training. In addition, we conduct controlled experiments by replacing the open-set dataset with a closed-set dataset to verify the effectiveness of open-set noisy dataset during the training. To demonstrate the importance of the sample size of the given auxiliary dataset, we compare the performance of training with different-sized auxiliary datasets, built by randomly sampling from the original auxiliary dataset.

Throughout this subsection, we perform training with WRN-40-2 [77] on CIFAR-10 [27] using standard cross-entropy loss. For experiments under inherent label noise, we use symmetric noise with 40% noisy rate. For auxiliary dataset, we use 300K Random Images[2] [20] as the open-set auxiliary dataset and CIFAR-5m[3] [21, 43] as the closed-set auxiliary dataset. Specifically, all images that belong to CIFAR classes are removed in 300K Random Images so that $\mathcal{D}_{\text{train}}$ and $\mathcal{D}_{\text{out}}$ are disjoint. More training details can be found in Appendix B.

**Learning with clean labels.** Fig. 1a presents the results on clean training set of CIFAR-10, training with the open/closed-set noisy auxiliary dataset. The results show that open-set noisy labels from the auxiliary dataset become harmless gradually with the increase of the sample size of the auxiliary dataset, while closed-set noisy labels consistently deteriorate the generalization performance. Furthermore, we observe that closed-set noisy labels result in unstable training shown by the large standard deviations in the figure. It is worthy to note that open-set noisy labels do not encounter this issue.

**Learning with inherent noisy labels.** Fig. 1b presents the results on CIFAR-10 with inherent noisy labels, training with the open/closed-set noisy auxiliary dataset. Surprisingly, we can observe that as we increase the sample size of the open-set noisy dataset, the open-set noises can even improve the generalization performance beyond standard training without the noisy auxiliary dataset. In this case, closed-set noisy auxiliary dataset still plays a negative role in the training.

**An intuitive interpretation**. From the results, one can hypothesize that open-set noisy labels in the auxiliary dataset would be harmless and can even benefit the robustness against inherent noisy labels, if the sample size of the auxiliary dataset is large enough. This phenomenon is somewhat counter-intuitive since one would expect that open-set noisy labels should have been "poison" to the training of network, like closed-set noises [30, 54, 61, 67]. Here we provide an intuitive interpretation from "insufficient capacity" [2]: increasing the number of examples while keeping representation capacity fixed would increase the time needed to memorize the data set. Hence, the larger the size of auxiliary dataset is, the more time it needs to memorize the open-set noises in the auxiliary dataset as

---

[2]The dataset is published on `https://github.com/hendrycks/outlier-exposure`.
[3]The dataset is published on `https://github.com/preetum/cifar5m`.

well as the inherent noises in the training set, relative to clean data. This interpretation is supported by our plot of training losses of inherent noises with different-sized auxiliary datasets in Fig. 1c, where increasing the number of open-set auxiliary samples slows down the fitting of inherent noises.

# 3    Method: Open-set regularization with Dynamic Noisy Labels

Motivated by the previous observations, we propose to utilize open-set auxiliary dataset to further improve the robustness against inherent noisy labels. This approach can be considered as an auxiliary task that continuously consumes the extra capacity of neural networks but does not interfere with learning patterns from clean data. Such auxiliary tasks will prevent deep networks from overfitting noisy data.

**Dynamic Noisy Labels.** To maintain the effectiveness of the auxiliary task during training, an ideal method is to use an auxiliary dataset with unlimited instances based on "insufficient capacity" [2]. In this way, the network would always try to fit a stream of new instances sequentially in an online learning setting. However, it is unrealistic to provide such an auxiliary dataset in real scenarios.

To relieve the requirement on sample size, we propose an alternative approach called Dynamic Noisy Labels. For each open-set instance $\widetilde{x}_i$, a noisy label $\widetilde{y}_i$ is uniformly sampled from the label set $\{1, \ldots, k\}$ independently, regardless of the instance $\widetilde{x}_i$. More importantly, the generated labels are not fixed and would be changed dynamically over training epochs. In this way, the auxiliary task becomes an "impossible mission", continuously consuming the extra capacity of network. Without Dynamic Noisy Labels, the effectiveness of the auxiliary task would be degraded gradually during training, because deep network can easily memorize the finite open-set samples, even though the labels of those samples are randomly generated [2, 79].

**Training Objective.** For the original training dataset, we use the standard cross entropy as the training objective function:

$$\mathcal{L}_1 = \mathbb{E}_{\mathcal{D}_{\text{train}}} \left[ \ell \left( f(\boldsymbol{x}; \boldsymbol{\theta}), y \right) \right] = \mathbb{E}_{\mathcal{D}_{\text{train}}} \left[ -\boldsymbol{e}^y \log f \left( \boldsymbol{x}; \boldsymbol{\theta} \right) \right], \tag{2}$$

For the auxiliary dataset, we enforce the outputs to be consistent with dynamic noise labels, uniformly sampled from the label set in each epoch. Therefore, the training objective function is as follows:

$$\mathcal{L}_2 = \mathbb{E}_{\mathcal{D}_{\text{out}}} \left[ \ell \left( f(\widetilde{\boldsymbol{x}}; \boldsymbol{\theta}), \widetilde{y} \right) \right] = \mathbb{E}_{\mathcal{D}_{\text{out}}} \left[ -\boldsymbol{e}^{\widetilde{y}} \log f \left( \widetilde{\boldsymbol{x}}; \boldsymbol{\theta} \right) \right], \text{where } \widetilde{y} \sim \mathcal{U}_k \tag{3}$$

where $\mathcal{U}_k$ represents a discrete uniform distribution on the label set $\{1, \ldots, k\}$.

Combining the original training dataset and the auxiliary dataset in the training, now we can formalize ODNL as minimizing the final objective:

$$\mathcal{L}_{\text{total}} = \mathcal{L}_1 + \eta \cdot \mathcal{L}_2 = \mathbb{E}_{\mathcal{D}_{\text{train}}} \left[ \ell \left( f(\boldsymbol{x}; \boldsymbol{\theta}), y \right) \right] + \eta \cdot \mathbb{E}_{\mathcal{D}_{\text{out}}} \left[ \ell \left( f(\widetilde{\boldsymbol{x}}; \boldsymbol{\theta}), \widetilde{y} \right) \right] \tag{4}$$

where $\eta$ denotes the trade-off hyperparameter. The details of ODNL are provided in Algorithm 1.

**Extensions to Other Robust Training Algorithms.** It is worthy to note that our regularization is a general method and can be easily incorporated into existing robust training algorithms, including sample selection[15, 63], loss correction [50], robust loss function [80, 62], etc. Given the original learning objective $\mathcal{L}_{\text{robust}}$ of the robust methods and the hyperparameter $\eta$, we can formalize the final objective as:

$$\mathcal{L}_{\text{total}} = \mathcal{L}_{\text{robust}} + \eta \cdot \mathcal{L}_2 \tag{5}$$

# 4    Understanding from SGD noise

To further understand the effect of our regularization, we provide an analysis from the lens of SGD noise. Following SLN [7], here we consider an original training sample $(\boldsymbol{x}, y)$ and an open-set sample $\widetilde{x}$ at each SGD step for convenience. In parameter updates, our regularization adds noises to the original gradients with the open-set noisy sample $\widetilde{x}$ and dynamic noisy label $\widetilde{y} \sim \mathcal{U}_k$:

$$\widetilde{\nabla}_{\boldsymbol{\theta}} \ell_{\text{total}} = \nabla_{\boldsymbol{\theta}} \ell(f(\boldsymbol{x}; \boldsymbol{\theta}), y) + \eta \cdot \nabla_{\boldsymbol{\theta}} \ell(f(\widetilde{\boldsymbol{x}}; \boldsymbol{\theta}), \widetilde{y}) \tag{6}$$

---

**Algorithm 1** Open-set Regularization with Dynamic Noisy Labels

---

**Input:** Training dataset $\mathcal{D}_{\text{train}}$. Open-set auxiliary dataset $\mathcal{D}_{\text{out}}$;
 1: **for** each iteration **do**
 2:    Sample a mini-batch of original training data $\{(\boldsymbol{x}_i, y_i)\}_{i=0}^{n}$ from $\mathcal{D}_{\text{train}}$;
 3:    Sample a mini-batch of open-set data $\{\widetilde{\boldsymbol{x}}_i\}_{i=0}^{m}$ from $\mathcal{D}_{\text{out}}$;
 4:    Generate random noisy label $\widetilde{y}_i \sim \mathcal{U}_k$ for each open-set data $\widetilde{\boldsymbol{x}}_i$;
 5:    Perform gradient descent on $f$ with $\mathcal{L}_{\text{total}}$ from Equation (4);
 6: **end for**

---

where $\ell$ denotes the cross entropy loss for single sample. For convenience, we omit the hyperparameter $\eta$ and use $\boldsymbol{f}$, $\widetilde{\boldsymbol{f}}$ to indicate the model output $f(\boldsymbol{x}; \boldsymbol{\theta})$ for the original training sample and the model output $f(\widetilde{\boldsymbol{x}}; \boldsymbol{\theta})$ for the open-set noisy sample. Following the standard notation of the Jacobian matrix, we have $\nabla_{\boldsymbol{\theta}}\ell \in \mathbb{R}^{1\times p}, \nabla_{\boldsymbol{f}}\ell \in \mathbb{R}^{1\times k}, \nabla_{\boldsymbol{\theta}}\boldsymbol{f} \in \mathbb{R}^{k\times p}, \nabla_{\boldsymbol{\theta}_i}\boldsymbol{f} \in \mathbb{R}^{k}$ and $\nabla_{\boldsymbol{\theta}}\boldsymbol{f}_i \in \mathbb{R}^{1\times p}$. The proofs of the following propositions are presented in Appendix A.

**Proposition 1.** *For the cross-entropy loss, Eq. (6) induces noise $\boldsymbol{z} = -\frac{\nabla_{\boldsymbol{\theta}}\widetilde{\boldsymbol{f}}_j}{\widetilde{\boldsymbol{f}}_j}$ on $\nabla_{\boldsymbol{\theta}}\ell(\boldsymbol{f}, y)$, s.t., $\boldsymbol{z} \in \mathbb{R}^p, j \sim \mathcal{U}_k$, where $\frac{\cdot}{\widetilde{\boldsymbol{f}}_j}$ denotes the element-wise division. Note that the expected value of noise on the $i$-th parameter $\boldsymbol{\theta}_i$ is $-\frac{1}{k}\sum_{j=1}^{k}\frac{\nabla_{\boldsymbol{\theta}_i}\widetilde{\boldsymbol{f}}_j}{\widetilde{\boldsymbol{f}}_j}$.*

**The effects of SGD noise.** From Proposition 1, we find that our regularization induces SGD noise sampled from a uniform distribution over $\{-\nabla_{\boldsymbol{\theta}}\widetilde{\boldsymbol{f}}_j/\widetilde{\boldsymbol{f}}_j\}_{j=1}^{k}$. The effects of the noises can be analyzed from three aspects:

- Firstly, our regularization yields SGD noises with random direction, uniformly sampled from the gradients of the open-set noisy sample. In this way, the SGD noises make the training difficult to converge and help escape local minima [4, 22, 25, 26, 46]. Without the noises, the model would quickly converge to a local minimum because the optimization of parameters always follows the direction of gradient descent during the training.

- Secondly, the noises $z$ are independently random to the gradients $\nabla_{\boldsymbol{\theta}}\ell(\boldsymbol{f}, y)$ from the original training example as the noises are conducted with the open-set sample $\widetilde{\boldsymbol{x}}$ from $\mathcal{D}_{out}$. If closed-set samples are used here, the generated noises might be conflicted with the original gradients, leading to unstable training and limiting the improvement from noises. This argument is clearly supported by the experimental results in Fig. 1 and Fig. 3a.

- Finally, the expected value of noise is not zero so the noise is biased. In particular, the expected value of noise enforces the model $f$ to produce more conservative predictions on Out-of-Distribution samples. Consequently, this feature would lead to performance improvement on OOD detection tasks as shown in Subsection 5.4.

To further demonstrate the importance of open-set samples in the auxiliary dataset, we conduct experiments with synthetic auxiliary datasets consisting of convex combinations $\boldsymbol{x}'$ of open-set samples $\widetilde{\boldsymbol{x}}$ and closed-set samples $\boldsymbol{x}$ with a hyperparameter $\alpha$: $\boldsymbol{x}' = (1-\alpha)\cdot\widetilde{\boldsymbol{x}} + \alpha\cdot\boldsymbol{x}$. Here, we use 300K Random Images [20] and CIFAR-5m [21, 43] as the open-set and closed-set dataset. More training details can be found in Appendix B. The results in Fig. 3a show that using open-set auxiliary dataset achieves better performance on robustness against inherent noises.

**Relations to label randomization methods.** Label randomization is also applied in some existing work in the context of deep learning[7, 36, 71]. For example, DisturbLabel [71] intentionally generates incorrect labels on a small fraction of training data to improve generalization performance under the settings of clean labels, interpreted as an implicit model ensemble. It can be seen as using a part of training data as a closed-set auxiliary dataset. In such a way, DisturbLabel can be interpreted as a special case of our method with a closed-set auxiliary dataset.

To mitigate the issue of label noise, Stochastic Label Noise (SLN) [7] introduces a variant of SGD noise induced by adding dynamic and mean-zero perturbations on the labels of training dataset, while Label Smoothing [36] uses a fixed and biased perturbation on the label. The noise gradients induced

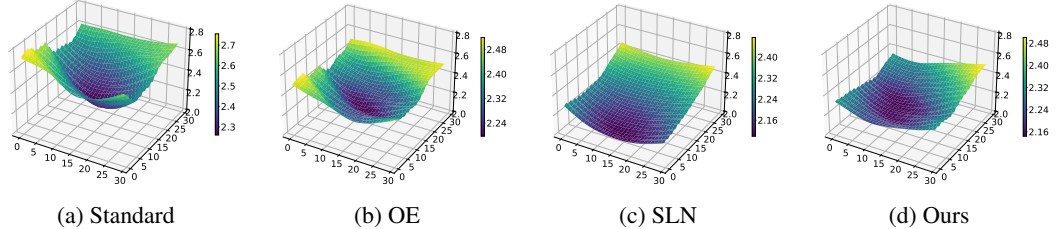

|     |     |     |     |
| :-: | :-: | :-: | :-: |
| (a) Standard | (b) OE | (c) SLN | (d) Ours |

Figure 2: Loss landscapes around the local minimum of models trained on CIFAR-10 with symmetric-40% noisy labels. We compare their sharpness with the same-scale z-axis and show the loss distribution by drawing color bars separately. We show that our method and SLN lead to a flat minimum, while the models trained by Standard and OE converge to a sharp minimum.

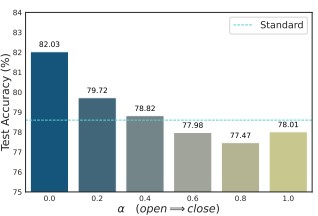 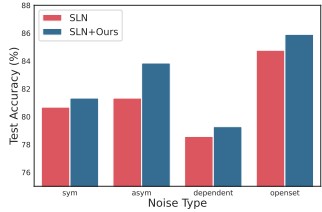 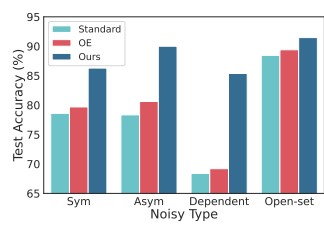

|     |     |     |
| :-: | :-: | :-: |
| (a) The importance of open set | (b) Our method can improve SLN | (c) Compare with OE. |

Figure 3: (a) Test accuracy (%) of ODNL ($\eta = 1$) on CIFAR-10 with symmetric-40% inherent noisy labels using different synthetic auxiliary datasets: the larger the $\alpha$ is, the closer the auxiliary sample distribution $\mathcal{D}_{\text{out}}$ is to the origin sample distribution $\mathcal{D}_{\text{train}}$. (b) Comparison of performances on CIFAR-10 with different types of inherent noisy labels using SLN and SLN+Ours. Here we follow the settings of SLN and the training details are provided in Appendix B. (c) Comparison of performances on CIFAR10 with different types of inherent noisy labels using OE and Ours.

by SLN are given as follows:

$$\tilde{\nabla}_{\boldsymbol{\theta}}\ell(\boldsymbol{f}, y) = \nabla_{\boldsymbol{\theta}}\ell\left(\boldsymbol{f}, \boldsymbol{e}^y + \sigma_y \boldsymbol{z}_y\right) \tag{7}$$

where $\sigma_y > 0, \boldsymbol{z}_y \in \mathcal{R}^k, \boldsymbol{z}_y \sim \mathcal{N}\left(0, \boldsymbol{I}_{k \times k}\right)$.

**Proposition 2** (Proposition 3 in Chen et al. [7]). *For the cross-entropy loss, Eq. (7) in SLN induces noise $\boldsymbol{z} \sim \mathcal{N}(0, \sigma_y^2 M)$ on $\nabla_{\boldsymbol{\theta}}\ell(\boldsymbol{f}, y)$, s.t., $\boldsymbol{M} \in \mathbb{R}^{p \times p}$ and $\boldsymbol{M}_{i,j} = \left(\frac{\nabla_{\boldsymbol{\theta}_i}\boldsymbol{f}}{\boldsymbol{f}}\right)^T \frac{\nabla_{\boldsymbol{\theta}_j}\boldsymbol{f}}{\boldsymbol{f}}, \forall i,j \in \{1, \ldots, p\}$, where $\frac{\cdot}{\boldsymbol{f}_j}$ denotes the element-wise division.*

From Proposition 2, we show that SLN induces SGD noises that obey a Gaussian distribution, where the standard deviation can be adjusted by the hyperparameter $\sigma_y$. Therefore, SLN and our ODNL can be considered as different variants of SGD noise, helping the network converge to a flat minimum, which is expected to provide better generalization performance [4, 22, 25, 26, 46]. The effects are explicitly presented in Fig. 2c and Fig. 2d. Moreover, the results in Fig. 3b illustrate that our method could further improve the performance of SLN[4] under various types of label noise, showing the complementarity between SLN and our method.

**Relations to Outlier Exposure.** Outlier Exposure (OE) [20] is an effective method designed for OOD detection tasks. Specifically, OE also makes use of a large, unlabeled auxiliary dataset sampled from $\mathcal{D}_{\text{out}}$, and regularizes the softmax probabilities to be a uniform distribution for out-of-distribution data in the auxiliary dataset. In such manner, OE train the model to discover signals and learn heuristics to detect whether a query is sampled from $\mathcal{D}_{\text{train}}$ or $\mathcal{D}_{\text{out}}$. The training objective is to minimize: $\mathbb{E}_{(\boldsymbol{x},y)\sim\mathcal{D}_{\text{train}}}\left[\ell(f(\boldsymbol{x};\boldsymbol{\theta}),y)\right] + \lambda \cdot \mathbb{E}_{\widetilde{\boldsymbol{x}}\sim\mathcal{D}_{\text{out}}}\left[-\frac{1}{k}\cdot\sum_{i=1}^{k}\log f_i(\widetilde{\boldsymbol{x}};\boldsymbol{\theta})\right]$.

**Proposition 3.** *For the cross-entropy loss, each open-set sample in OE induces bias $\boldsymbol{z} = -\frac{1}{k}\sum_{j=1}^{k}\frac{\nabla_{\boldsymbol{\theta}}\widetilde{\boldsymbol{f}}_j}{\widehat{\boldsymbol{f}}_j}$ on $\nabla_{\boldsymbol{\theta}}\ell(\boldsymbol{f}, y)$, s.t., $\boldsymbol{z} \in \mathbb{R}^p$, where $\frac{\cdot}{\widehat{\boldsymbol{f}}_j}$ denotes the element-wise division.*

---

[4]We use the official implementation: https://github.com/chenpf1025/SLN.

Table 1: Average test accuracy (%) with standard deviation on CIFAR-10 under various types of noisy labels (over 5 trials). The bold indicates the improved results by integrating our regularization.

| Method | Sym-20% | Sym-50% | Asymmetric | Dependent | Open-set |
|---|---|---|---|---|---|
| Standard | 86.93±0.49 | 69.48±0.58 | 78.34±0.96 | 68.39±0.65 | 88.45±0.57 |
| + ODNL (Ours) | **91.06±0.64** | **82.50±0.58** | **90.00±0.35** | **85.37±0.32** | **91.47±0.39** |
| Decoupling | 88.31±0.39 | 80.81±0.85 | 84.85±0.22 | 82.44±0.54 | 84.69±0.46 |
| + ODNL (Ours) | **89.70 ±0.74** | **81.48±0.37** | **87.36±0.68** | **84.11±0.51** | **86.22±0.57** |
| F-correction | 84.95±0.27 | 72.11±0.15 | 84.46±0.96 | 72.44±0.58 | 85.54±0.86 |
| + ODNL (Ours) | **88.99±0.23** | **82.42±0.29** | **88.50±0.15** | **84.70±0.39** | **88.92±0.65** |
| PHuber-CE | 90.92±0.28 | 74.07±0.49 | 81.26±0.65 | 75.07±1.21 | 88.35±0.21 |
| + ODNL (Ours) | **92.05±0.36** | **82.63±0.28** | **88.24±0.43** | **84.11±0.45** | **90.85±0.29** |
| Co-teaching | 91.72±0.35 | 86.65±0.43 | 87.56±0.36 | 88.64±0.79 | 89.91±0.58 |
| + ODNL (Ours) | **93.07±0.79** | **89.79±0.76** | **88.48±0.37** | **90.99±0.28** | **91.88±0.53** |
| JoCoR ($\lambda = 0.5$) | 92.68±0.33 | 88.36±0.64 | 84.06±0.37 | 90.11±0.19 | 88.80±0.54 |
| + ODNL (Ours) | **93.85±0.69** | **91.17±0.52** | **88.10±0.69** | **92.09±0.84** | **91.73±0.73** |

Comparing Proposition 3 with Proposition 1, our method should have similar regularization effect to OE on OOD examples, because the expectation of the SGD noises from ODNL is equal to the bias induced by OE. However, OE cannot improve the robustness against noisy labels as it does not induce dynamic noises so that the optimization of parameters always follows the direction of gradient descent. The difference is empirically shown in Fig. 3c and we further verify it in Subsection 5.4.

## 5 Experiments

In this section, we verify the effectiveness of our method on three benchmark datasets: CIFAR-10/100 and Clothing1M. We show that ODNL could not only significantly improve the robustness against various types of inherent noisy labels in standard training, but also enhance the performance of existing robust training techniques. Then we perform a sensitivity analysis to validate the effect of $\eta$. Finally, we conduct experiments to evaluate the performance of ODNL on OOD detection task.

### 5.1 Setups

We comprehensively verify the utility of ODNL[5] on different types of label noise, including symmetric noise, asymmetric noise[80], instance-dependent noise [6] and open-set noise [61] synthesized on CIFAR-10/100 and real-world noise on Clothing1M [70]. Symmetric noise assumes each label has the same probability of flipping to any other classes. We uniformly flip the label to other classes with an overall probability 20% and 50%. Asymmetric noise assumes labels might be only flipped to similar classes [7, 50, 80]. Noisy labels are generated by mapping TRUCK → AUTOMOBILE, BIRD → AIRPLANE, DEER → HORSE, and CAT ↔ DOG with probability 40% for CIFAR-10. For CIFAR-100, we flip each class into the next circularly with probability 40%. Instance-dependent noise assumes the mislabeling probability is dependent on each instance's input features [6, 7, 68]. We use the instance-dependent noise from PDN [68] with a noisy rate 40%, where the noise is synthesized based on the DNN prediction error. Open-set noise contains samples that do not belong to the label set considered in the classification task [61]. We generate open-set noises on CIFAR-10 by randomly replacing 40% of its training images with images from CIFAR-100.

We perform training with WRN-40-2 [77] on CIFAR-10 and CIFAR-100. The network is trained for 200 epochs using SGD with a momentum of 0.9, a weight decay of 0.0005. In each iteration, both the batch sizes of the original dataset and the open-set auxiliary dataset are set as 128. We set the initial learning rate as 0.1, and reduce it by a factor of 10 after 80 and 140 epochs. We use 5k noisy samples as the validation to tune the hyperparameter $\eta$ in $\{0.1, 0.5, 1, 2.5, 5\}$, then train the model on the full training set and report the average test accuracy in the last 5 epochs. All experiments are repeated five times with different seeds and more training details can be found in Appendix B. We use 300K Random Images [20] as the open-set auxiliary dataset and more discussions on the different choices of open-set auxiliary dataset are presented in Appendix C.

---

[5]The code is published on https://github.com/hongxin001/ODNL

Table 2: Average test accuracy (%) with standard deviation on CIFAR-100 under various types of noisy labels (over 5 trials). The bold indicates the improved results by integrating our regularization.

| Method | Sym-20% | Sym-50% | Asymmetric | Dependent |
|---|---|---|---|---|
| Standard | 64.87±0.81 | 48.55±0.65 | 48.77±0.86 | 53.94±0.52 |
| **+ ODNL (Ours)** | **68.82±0.24** | **54.08±0.54** | **58.61±0.25** | **62.45±0.41** |
| Decoupling | 64.26±0.23 | 49.09±0.81 | 49.92±0.71 | 55.17±0.34 |
| **+ ODNL (Ours)** | **66.86±0.35** | **51.89±0.47** | **57.47±0.49** | **58.52±0.28** |
| F-correction | 62.05±0.35 | 52.27±0.15 | 54.45±0.94 | 57.47±0.26 |
| **+ ODNL (Ours)** | **65.53±0.28** | **57.12±0.65** | **57.89±0.55** | **60.95±0.81** |
| PHuber-CE | 71.27±0.74 | 61.37±0.28 | 48.14±0.35 | 64.80±0.32 |
| **+ ODNL (Ours)** | **72.27±0.53** | **63.33±0.48** | **53.63±0.72** | **66.54±0.60** |
| Co-teaching | 72.35±0.60 | 65.21±0.88 | 54.16±0.17 | 67.85±0.82 |
| **+ ODNL (Ours)** | **73.12±0.20** | **65.46±0.49** | **54.92±0.31** | **68.37±0.12** |
| JoCoR ($\lambda = 0.5$) | 73.44±0.19 | 67.53±0.79 | 57.02±0.32 | 70.01±0.26 |
| **+ ODNL (Ours)** | **74.48±0.16** | **68.07±0.47** | **58.44±0.34** | **71.28±0.31** |

Table 3: Test accuracy (%) on Clothing1M with pretrained ResNet-50. All experiments are implemented based on DivideMix's official implementation. The bold indicates the improved results.

| Method | Standard | Forward | Co-teaching | **ODNL (Ours)** | SLN | **+ ODNL (Ours)** | DivideMix | **+ ODNL (Ours)** |
|---|---|---|---|---|---|---|---|---|
| *best* | 70.17 | 70.46 | 71.32 | **72.47** | 72.31 | **73.12** | 74.32 | **74.89** |
| *last* | 68.23 | 69.34 | 71.02 | **72.38** | 71.94 | **72.98** | 73.83 | **74.35** |

## 5.2 CIFAR-10 and CIFAR-100

On CIFAR-10 and CIFAR-100, we verify that ODNL can boost existing robust training methods by integrating ODNL with the following methods: 1) Decoupling [40], which updates the parameters only using instances that have different predictions from two classifiers. 2) F-correction [50], which corrects the prediction by the label transition matrix. As the setting in F-correction, we first train a standard network to estimate the transition matrix Q. 3) PHuber-CE [41], which uses a composite loss-based gradient clipping, a variation of standard gradient clipping for label noise robustness. 4) Co-teaching [15], which trains two networks simultaneously and cross-updates parameters of peer networks. 5) JoCoR [63], which trains two networks simultaneously and reduces the divergence between the two classifiers by explicit regularization.

The average test accuracy with WRN-40-2 [77] are reported in Table 1 and Table 2. The results show that incorporating our regularization into existing robust training methods consistently improves their performance against various types of noisy labels. In particular, we can observe that our method can boost almost all types of existing methods, including robust loss function, sample selection, loss correction, etc., showing that our regularization based upon an open-set auxiliary dataset is a complementary method to existing robust algorithms. More intriguingly, while most existing methods do not bring benefits to the robustness against open-set noisy labels in the training set of CIFAR-10, our regularization could still enhance the performance of all existing methods in this case.

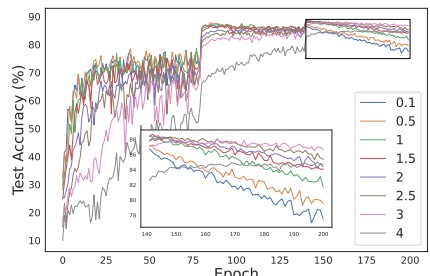

Figure 4: Results of sensitivity analysis on CIFAR-10 with different $\eta$.

To further illustrate the influence of $\eta$, we present a sensitivity analysis on CIFAR-10 with symmetric-40% noisy labels in Fig. 4. Specifically, we highlight the differences in the trend of test accuracy after the second decay of learning rate at the 140th epoch. We can observe that with a large $\eta$, the decreasing trend of test accuracy caused by inherent noisy labels is nearly eliminated. The result verifies that our regularization is an effective method to reduce the overfitting of inherent noisy labels.

Table 4: OOD detection performance comparison on CIFAR-10 with symmetric-40% noisy labels. All values are percentages and are averaged over the ten test datasets described in Appendix B. ↑ indicates larger values are better, and ↓ indicates smaller values are better. Bold numbers are superior results. Detailed results for each OOD test dataset can be found in Appendix C

| Method | Test Accuracy ↑ | FPR95 ↓ | AUROC ↑ | AUPR ↑ |
|---|---|---|---|---|
| MSP | 77.87 | 92.01 | 54.64 | 19.41 |
| MSP+OE | 78.12 | 24.91 | 94.24 | 78.09 |
| MSP+Ours | **86.29** | **13.43** | **96.67** | **84.70** |

### 5.3 Clothing1M

Clothing1M [70] is a large-scale dataset with real-world noisy labels of clothing images. There are 1 million noisy samples for training, 14K and 10K clean examples for validation and test respectively. Following DivideMix [33], we conduct experiments by sampling a class-balanced training subset in each epoch and use ResNet-50 with ImageNet pretrained weights. We set the $\eta$ of the vanilla ODNL as 5. For the auxiliary dataset in our method, we use ImageNet 2012 dataset [53] and remove all classes related to clothes. Other training details strictly follow the settings of DivideMix [33], presented in Appendix B. As shown in Table 3, *best* denotes the score of the epoch where the validation accuracy is optimal, and *last* denotes the scores at the last epoch. The vanilla ODNL outperforms many simple baselines, and it can further improve the performance of SLN and DivideMix. The results show that our method is applicable for large-scale real-world scenarios.

### 5.4 OOD detection with noisy labels

Out-of-distribution (OOD) detection is an essential problem for the deployment of deep learning especially in safety-critical applications [20, 35]. From the analysis in Section 4, we know that our regularization should have a similar regularization effect to OE [20], which enforces the model to give conservative outputs on OOD examples. Here, we conduct experiments on CIFAR-10 with symmetric-40% noisy labels to verify the advantage of our method on OOD detection tasks even in a weakly supervised setting. We expect the trained models could perform well in the OOD detection task, even if the labels of training dataset are noisy. Experiments on CIFAR-10 with clean labels are also conducted and the results can be found in Appendix C. The training settings are the same as those in subsection 5.2. Following OE [20], we consider the maximum softmax probability (MSP) baseline [18], which gives the maximum of softmax output as the OOD score. For the OOD test datasets, we use three types of noises and seven common benchmark datasets: Gaussian, Rademacher [20], Blobs [20], Textures [9], SVHN [45], CIFAR100, Places365 [82], LSUN-Crop [75], LSUN-Resize [75], and iSUN [72]. We measure the following metrics: (1) the false positive rate (FPR95) of OOD examples when the true positive rate of in-distribution examples is at 95%; (2) the area under the receiver operating characteristic curve (AUROC); and (3) the area under the precision-recall curve (AUPR). Table 4 presents the test accuracy and the average performance over the three types of noises and seven OOD test datasets. We can observe that our method achieves impressive improvement on both the test accuracy and the detection performance, while OE does not bring benefit on the generalization performance. The results can further serve as evidence for the analysis in Section 4.

## 6 Related literature

**Learning with noisy labels.** Learning with noisy labels is a vital topic in weakly-supervised learning, attracting much recent interest with several directions. 1) Some methods propose to train on selected samples, using small-loss selection [15, 63, 76], GMM distribution [1, 33] or (dis)agreement between two models [40, 63, 76]. 2) Some methods aim to design sample weighting schemes that give higher weights on clean samples [24, 34, 52, 55, 64]. 3) Loss correction is also a popular direction based on an estimated noise transition matrix [19, 50] or the model's predictions [1, 6, 51, 58, 81]. 4) Robust loss functions are also studied to have a theoretical guarantee [11, 12, 37, 38, 62, 73, 80]. 5) Some method apply regularization techniques to improve generalization under the settings of label noise [10, 23, 66], like gradient clipping [41], label smoothing [36, 57], temporal ensembling [28] and virtual adversarial training [42]. 6) Some training strategies for combating noisy labels are built based

upon semi-supervised learning methods [33, 47]. Studying robustness against label noise helps us further understand the essence of training deep neural networks.

**Utilizing auxiliary dataset.** To the best of our knowledge, we are the first to explore the benefits of open-set auxiliary dataset in learning with noisy labels. Auxiliary dataset is also utilized in other problem settings of deep learning. For example, pre-training a network on the large ImageNet database [53] can produce general representations that are useful in many fine-tuning applications [78]. Representation learned from images scraped from search engines and photo-sharing websites could improve objective detection performance [8, 39]. OE uses an auxiliary dataset to teach the network better representations for OOD detection [20]. To improve robustness against adversarial attack, a popular method is to train on adversarial examples which can be seen as a generated auxiliary dataset [14]. Unlabelled data is also shown to be beneficial for the adversarial robustness [5, 59]. Note that the unlabelled data for improving adversarial robustness is required to be in-distribution, our method utilizes out-of-distribution instances to improve robustness against noisy labels. Recently, OAT [31] designed to use out-of-distribution data for improving robustness against adversarial examples, a detailed comparison between OAT and our work is provided in Appendix D.

**OOD detection.** OOD detection is an essential building block for safely deploying machine learning models in the open world [18, 20, 35]. A common baseline for OOD detection is the softmax confidence score [18]. It has been theoretically shown that neural networks with ReLU activation can produce arbitrarily high softmax confidence for OOD inputs [17]. To improve the performance, previous methods have explored using artificially synthesized data from GANs [13] or unlabeled data [20, 29] as auxiliary OOD training data. Energy scores are shown to be better for distinguishing in- and out-of-distribution samples than softmax scores [35]. As a side effect, our method could also achieve superior performance in OOD detection tasks even if the labels of training dataset are noisy.

# 7 Conclusion and Discussion

In this paper, we proposed Open-set regularization with Dynamic Noisy Labels (ODNL), a simple yet effective technique that enhances many existing robust training methods to mitigate the issues of inherent noisy labels across various settings. To the best of our knowledge, our method is the first to utilize open-set auxiliary dataset in the problem of noisy labels. We show that ODNL can be considered as a variant of SGD noises that usually leads to a flat minimum. Extensive experiments show that ODNL can help improve the robustness against various types of noisy labels and it is applicable for large-scale real-world scenarios. Moreover, ODNL also brings a "side effect": it can improve the OOD detection performance, which is also essential for the deployment of deep learning in safety-critical applications. On the other hand, ODNL is computationally inexpensive and can be easily integrated into existing algorithms. Overall, our method is an effective and complementary approach for boosting robustness against inherent noisy labels.

In the future, the observations and analyses in this work can help understand the effect of open-set noises and inspire more specially designed methods using auxiliary dataset to combat label noise in deep learning. There are still some limitations in our current exploration and method as follows.

*instance-dependent open-set noisy labels.* In this work, we only consider open-set noisy labels that are independent from the instances. In real scenarios, an open-set sample with a noisy label might be similar to the clean samples in this class but still do not belong to this class. We may get different observations when we extend the setting to instance-dependent open-set noisy labels.

*Synthesized auxiliary dataset.* In this work, we use realistic open-set dataset in our method. It is also natural to ask whether synthesized data can be used to combating noisy labels. For example, one may generate adversarial examples with dynamic noisy labels to improve robustness against noisy labels.

## Acknowledgments and Disclosure of Funding

This research was supported by the National Research Foundation, Singapore under its AI Singapore Programme (AISG Award No: AISG-RP-2019-0013), National Satellite of Excellence in Trustworthy Software Systems (Award No: NSOE-TSS2019-01), and NTU.

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
