# Appendices

## A  Proofs

**Proposition 4.** *For the cross-entropy loss, Eq. (6) induces noise $z = -\frac{\nabla_{\boldsymbol{\theta}} \widetilde{\boldsymbol{f}}_j}{\widetilde{\boldsymbol{f}}_j}$ on $\nabla_{\boldsymbol{\theta}} \ell(\boldsymbol{f}, y)$, s.t., $z \in \mathbb{R}^p, j \sim \mathcal{U}_k$, where $\frac{\dot{}}{\widetilde{\boldsymbol{f}}_j}$ denotes the element-wise division. Note that the expected value of noise on the $i$-th parameter $\boldsymbol{\theta}_i$ is $-\frac{1}{k} \sum_{j=1}^{k} \frac{\nabla_{\boldsymbol{\theta}_i} \widetilde{\boldsymbol{f}}_j}{\widetilde{\boldsymbol{f}}_j}$.*

*Proof.* For Eq. (6), the noisy gradient is

$$\widetilde{\nabla}_{\boldsymbol{\theta}} \ell_{\text{total}} = \nabla_{\boldsymbol{\theta}} \ell(\boldsymbol{f}, y) + \eta \cdot \nabla_{\boldsymbol{\theta}} \ell\left(\widetilde{\boldsymbol{f}}, \widetilde{y}\right) = \nabla_{\boldsymbol{\theta}} \ell(\boldsymbol{f}, y) + \eta \cdot \nabla_{\widetilde{\boldsymbol{f}}} \ell\left(\widetilde{\boldsymbol{f}}, \widetilde{y}\right) \cdot \nabla_{\boldsymbol{\theta}} \widetilde{\boldsymbol{f}}$$

For convenience, we omit the hyperparameter $\eta$. Note that cross-entropy loss is $\ell(\boldsymbol{f}, y) = -e^y \log \boldsymbol{f}$, then the noise on $\nabla_{\boldsymbol{\theta}} \ell(\boldsymbol{f}, y)$ is

$$z = \nabla_{\widetilde{\boldsymbol{f}}} \ell\left(\widetilde{\boldsymbol{f}}, \widetilde{y}\right) \cdot \nabla_{\boldsymbol{\theta}} \widetilde{\boldsymbol{f}} = -\left(\frac{e^{\widetilde{y}}}{\widetilde{\boldsymbol{f}}}\right)^T \cdot \nabla_{\boldsymbol{\theta}} \widetilde{\boldsymbol{f}} = -\sum_{j=1}^{k}\left(e_j^{\widetilde{y}} \cdot \frac{\nabla_{\boldsymbol{\theta}_i} \widetilde{\boldsymbol{f}}_j}{\widetilde{\boldsymbol{f}}_j}\right), \quad \text{s.t. } \widetilde{y} \sim \mathcal{U}_k.$$

Since $e^{\widetilde{y}} = (0, \cdots, 1, \cdots, 0)$ is the one-hot vector and only the $y$-th entry is 1, the expression of the noise $z$ can be simplified as:

$$z = -\frac{\nabla_{\boldsymbol{\theta}} \widetilde{\boldsymbol{f}}_j}{\widetilde{\boldsymbol{f}}_j}, \quad \text{s.t. } j \sim \mathcal{U}_k.$$

Let $z_i$ be the $i$-th entry of $z$, we have

$$z_i = -\frac{\nabla_{\boldsymbol{\theta}_i} \widetilde{\boldsymbol{f}}_j}{\widetilde{\boldsymbol{f}}_j}, \quad \text{s.t. } j \sim \mathcal{U}_k.$$

Hence,

$$\mathbb{E}(z_i) = -\frac{1}{k} \sum_{j=1}^{k} \frac{\nabla_{\boldsymbol{\theta}_i} \widetilde{\boldsymbol{f}}_j}{\widetilde{\boldsymbol{f}}_j}$$

$\square$

**Proposition 3.** *For the cross-entropy loss, each open-set sample in OE induces bias $z = -\frac{1}{k} \sum_{j=1}^{k} \frac{\nabla_{\boldsymbol{\theta}} \widetilde{\boldsymbol{f}}_j}{\widetilde{\boldsymbol{f}}_j}$ on $\nabla_{\boldsymbol{\theta}} \ell(\boldsymbol{x}, y)$, s.t., $z \in \mathbb{R}^p$, where $\frac{\dot{}}{\widetilde{\boldsymbol{f}}_j}$ denotes the element-wise division.*

*Proof.* The regularization item of OE is $\ell_{\text{OE}} = -\frac{1}{k} \cdot \sum_{i=1}^{k} \log \widetilde{\boldsymbol{f}}_i$.
Then the gradient of $\ell_{\text{OE}}$ w.r.t $\theta$ is

$$\nabla_{\boldsymbol{\theta}} \ell_{\text{OE}} = \nabla_{\widetilde{\boldsymbol{f}}} \ell_{\text{OE}} \cdot \nabla_{\boldsymbol{\theta}} \widetilde{\boldsymbol{f}} = \nabla_{\widetilde{\boldsymbol{f}}}\left(-\frac{1}{k} \cdot \sum_{j=1}^{k} \log \widetilde{\boldsymbol{f}}_j\right) \cdot \nabla_{\boldsymbol{\theta}} \widetilde{\boldsymbol{f}} = -\frac{1}{k} \cdot \sum_{j=1}^{k} \frac{\nabla_{\boldsymbol{\theta}} \widetilde{\boldsymbol{f}}_j}{\widetilde{\boldsymbol{f}}_j}$$

$\square$

# B More details on experiment setup

## B.1 Datasets

**Auxiliary datasets**. 300K Random Images [20] is a cleaned and debiased dataset with 300K natural images. In this dataset, Images that belong to CIFAR classes from it, images that belong to Places or LSUN classes, and images with divisive metadata are removed so that $\mathcal{D}_{\text{train}}$ and $\mathcal{D}_{\text{out}}$ are disjoint. We use the dataset as the open-set auxiliary dataset for experiments with CIFAR-10 and CIFAR-100. In particular, For experiments with Clothing1M, ImageNet 2012 dataset [53] is used as the open-set auxiliary dataset and we remove all classes related to clothes. CIFAR-5m [21, 43] is a dataset of 6 million synthetic CIFAR-10-like images. For experiments in Fig. 1 and Fig. 3a, we build the closed-set dataset by sampling class-balanced subsets from CIFAR-5m.

**OOD test datasets**. Following OE [20], we comprehensively evaluate OOD detectors on artificial and real anomalous distributions, including: Gaussian, Rademacher, Blobs, Textures [9], SVHN [44], Places365 [82], LSUN-Crop [75], LSUN-Resize [75], iSUN [72]. For experiments on CIFAR-10, we also use CIFAR-100 as OOD test dataset. *Gaussian* noises have each dimension i.i.d. sampled from an isotropic Gaussian distribution. *Rademacher* noises are images where each dimension is $-1$ or $1$ with equal probability, so each dimension is sampled from a symmetric Rademacher distribution. *Blobs* noises consist in algorithmically generated amorphous shapes with definite edges. *Textures* [9] is a dataset of describable textural images. *SVHN* dataset [44] contains $32 \times 32$ color images of house numbers. There are ten classes comprised of the digits 0-9. *Places365* [82] consists in images for scene recognition rather than object recognition. *LSUN* [75] is another scene understanding dataset with fewer classes than Places365. Here we use *LSUN-Crop* and *LSUN-Resize* to denote the cropped and resized version of the LSUN dataset respectively. *iSUN* [72] is a large-scale eye tracking dataset, selected from natural scene images of the SUN database [69].

## B.2 Setups

We conduct all the experiments on NVIDIA GeForce RTX 3090, and implement all methods with default parameters by PyTorch [49]. Following GCE [80] and SLN [7], we use 5k noisy samples as the validation set to tune the hyperparameters. We then train the model on the full training set and report the average test accuracy over the last 5 epochs.

**For experiments in Fig. 1, Fig. 2, Fig. 3a and Fig. 3c**. We use WRN-40-2 [77] and the network trains for 200 epochs with a dropout rate of 0.3, using SGD with a momentum of 0.9, a weight decay of 0.0005. In each iteration, both the batch sizes of the original dataset and the open-set auxiliary dataset are set as 128. Note that we compare the performance of training with different-sized auxiliary datasets in Fig. 1, the batch size of the auxiliary dataset is fixed as 128 in all cases. We set the initial learning rate as 0.1, and reduce it by a factor of 10 after 80 and 140 epochs. For drawing loss landscapes in Fig. 2, we use the technique from the *loss-landscapes* library [32]. For experiments in Fig. 3c, we use 40% noise rate in all types of inherent noisy labels.

**For experiments in Fig. 3b**. We use the same setting as that of SLN [7]. Specifically, we train WRN-28-2 [77] for 300 epochs using SGD with learning rate 0.001, momentum 0.9, weight decay 0.0005 and a batch size of 128. For the hyperparameter $\sigma$ of SLN, we use $\sigma = 1$ for symmetric noise and $\sigma = 0.5$ otherwise. Here, we use 40% noise rate in all types of inherent noisy labels, following the official implementation of SLN.

**For experiments on Clothing1M**. We use the same setting as that of DivideMix [33]. Specifically, we use ResNet-50 with ImageNet pretrained weights for 80 epochs. The initial learning rate is set as 0.002 and reduced by a factor of 10 after 40 epochs. For each epoch, we sample 1000 mini-batches from the training data while ensuring the labels are balanced.

**Method-specific hyperparameters**. The backbone and training settings of the compared methods are not unified in previous paper and we reimplement all methods in the same backbone for a fair comparison. For experiments on CIFAR-10/100, we set $\lambda = 0.5$ for JoCoR [63]. For experiments on Clothing1M, we use the default hyperparameters for DivideMix: $M = 2, T = 0.5, \tau = 0.5, \lambda_u = 0, \alpha = 0.5$, and we set the standard deviation of SLN as $\sigma = 0.2$.

**Guideline for tuning** $\eta$. We tune the hyperparameter $\eta$ following a guideline that has been widely used in existing literature like SLN [7]. Given a new dataset with unknown noise, we suggest quickly

searching the best value of $\eta$ (denoted as $\eta_o$) based on the binary search using the validation accuracy throughout training. 1) If we observe a decrease of validation accuracy at the late stage of training, it implies overfitting and $\eta < \eta_o$. 2) Otherwise, we have $\eta \geq \eta_o$. Based on 1) and 2), we can conduct a binary search to quickly find the best value of $\eta$ even if one would like to search $\eta$ in a very detailed range. The best values of $\eta$ for vanilla ODNL are reported in Table 5.

Table 5: The best test accuracy (%) and the value of $\eta$ on CIFAR-10/100 using vanilla ODNL.

| Dataset | Method | Sym-20% | Sym-50% | Asym | Dependent | Open |
|---------|--------|---------|---------|------|-----------|------|
| CIFAR-10 | Ours | 91.06 | 82.50 | 90.00 | 85.37 | 91.47 |
| | $\eta$ | 2.5 | 2.5 | 3.0 | 3.5 | 2.0 |
| CIFAR-100 | Ours | 68.82 | 54.08 | 58.61 | 62.45 | 66.95 |
| | $\eta$ | 1.0 | 1.0 | 2.0 | 2.0 | 1.0 |

## C More empirical results

### C.1 Our method also works on other architectures.

Table 6: Test accuracy (%) on CIFAR-10 under symmetric-40% noise rate training with various architectures. The bold indicates the improved results.

| Method | WRN-40-2 | ResNet-18 | VGG-11 |
|--------|----------|-----------|--------|
| Standard | 77.55 | 58.60 | 58.94 |
| ODNL (Ours) | **86.29** | **80.88** | **74.27** |

Our regularization method improves robustness against label noise for more than just WRN architectures. Table 6 shows that ODNL also improves performance under various types of noisy labels with ResNet-18 [16] and VGG-11 [56]. In particular, we set weight decay as 0.001 for experiments with ResNet-18 and VGG-11.

### C.2 Different choices of open-set auxiliary datasets.

In the experiments in Subsection 2.2, we found that the sample size of auxiliary dataset is important for the generalization performance when the labels of open-set samples are fixed. To relieve this requirement, we introduce dynamic noisy labels to strengthen our method in Section 3. In particular, our method can use only 50,000 examples from 300K Random Images to achieve comparable performance with using all examples from this dataset. Moreover, we show that using simple Gaussian noises as auxiliary dataset can also improve robustness against noisy labels, but do not work as well as real data. For example, in the case of symmetric-40% noise on CIFAR-10, using ODNL with simple Gaussian noises achieves 80.24% accuracy while using ODNL with 300K Random Images achieve 82.03%. In addition to size and realism, we also find that diversity of auxiliary dataset is not important for robustness against noisy labels, while OE [20] claims that it is an important factor for OOD detection. We conduct experiments on CIFAR-10 with symmetric-40% noises and use the subset of CIFAR-100 with different number of classes as auxiliary dataset. The sample size of these subsets are fixed as 5,000. The results of using different subsets are very close, achieving 81.56% test accuracy.

### C.3 Detailed results for OOD detection.

Table 7 presents the detailed results of OOD detection performance on CIFAR-10 under symmetric-40% noisy labels with various OOD test datasets. From the results, we show that our method can outperform OE [20] in OOD detection tasks when the labels of training dataset are noisy. Table 8 presents the detailed results of OOD detection performance on CIFAR-10 under clean labels with various OOD test datasets. We can observe that our method achieve comparable performance to OE [20]. The results also support the analysis in Section 4.

Table 7: OOD detection performance comparison on CIFAR-10 under symmetric-40% noisy labels with different OOD test dataset. ↑ indicates larger values are better and ↓ indicates smaller values are better.

| OOD test dataset | Method | FPR95 ↓ | AUROC ↑ | AUPR ↑ |
|---|---|---|---|---|
| Gaussian | MSP | 66.14 | 72.29 | 27.08 |
| | MSP+OE | 6.58 | 97.39 | 78.37 |
| | MSP+Ours | 2.68 | 98.54 | 83.87 |
| Rademacher | MSP | 91.7 | 49.44 | 15.26 |
| | MSP+OE | 5.72 | 97.43 | 76.51 |
| | MSP+Ours | 1.19 | 99.34 | 91.30 |
| Blob | MSP | 99.4 | 28.05 | 10.92 |
| | MSP+OE | 12.51 | 97.42 | 90.62 |
| | MSP+Ours | 3.46 | 99.13 | 94.81 |
| Textures | MSP | 98.62 | 49.26 | 16.89 |
| | MSP+OE | 25.43 | 94.66 | 79.70 |
| | MSP+Ours | 11.64 | 97.43 | 88.67 |
| SVHN | MSP | 95.78 | 56.19 | 19.36 |
| | MSP+OE | 44.26 | 90.67 | 69.91 |
| | MSP+Ours | 18.53 | 95.38 | 79.63 |
| CIFAR-100 | MSP | 94.92 | 56.2 | 19.86 |
| | MSP+OE | 71.09 | 83.42 | 60.41 |
| | MSP+Ours | 50.16 | 88.50 | 66.55 |
| LSUN-C | MSP | 92.03 | 60.38 | 22.38 |
| | MSP+OE | 13.92 | 96.68 | 86.28 |
| | MSP+Ours | 9.47 | 97.52 | 87.47 |
| LSUN-R | MSP | 93.36 | 60.42 | 22 |
| | MSP+OE | 11.84 | 96.77 | 82.96 |
| | MSP+Ours | 6.16 | 98.11 | 87.37 |
| iSUN | MSP | 92.31 | 59.75 | 21.6 |
| | MSP+OE | 12.34 | 96.47 | 80.78 |
| | MSP+Ours | 6.50 | 98.01 | 86.53 |
| Places365 | MSP | 95.84 | 54.41 | 18.72 |
| | MSP+OE | 45.43 | 91.54 | 75.38 |
| | MSP+Ours | 24.49 | 94.75 | 80.81 |

# D  Discussion

**Relations to OAT.** OAT [31] aims to use out-of-distribution data to improve generalization in the context of adversarial robustness. While OAT is relevant to our work, we note that there are key differences in both technique and insight perspectives:

- Technique: OAT regularizes the softmax probabilities to be a uniform distribution for out-of-distribution data, which is the same as Outlier Exposure [20]. As analyzed in Section 4, the regularization in OE and OAT cannot improve the robustness against noisy labels, because it does not induce dynamic noises and the optimization of parameters always follows the direction of gradient descent.

- Insight: OAT shows that OOD data samples share the same undesirable features as those of the in-distribution data so that these samples could also be used to remove the influence of undesirable features in adversarial robustness. In our work, we show that open-set noisy labels could be harmless and even benefit the robustness against inherent noisy labels. We give an intuitive interpretation for the phenomena by "insufficient capacity" and understand the effects of open-set noises from the perspective of SGD noises.

Table 8: OOD detection performance comparison on CIFAR-10 with clean labels. ↑ indicates larger values are better and ↓ indicates smaller values are better.

| OOD test dataset | Method | FPR95 ↓ | AUROC ↑ | AUPR ↑ |
|---|---|---|---|---|
| Gaussian | MSP | 12.45 | 95.71 | 75.55 |
| | MSP+OE | 0.83 | 99.36 | 90.36 |
| | MSP+Ours | 0.74 | 99.59 | 94.87 |
| Rademacher | MSP | 23.68 | 87.46 | 40.80 |
| | MSP+OE | 0.94 | 97.69 | 87.82 |
| | MSP+Ours | 0.56 | 99.73 | 96.95 |
| Blob | MSP | 27.09 | 90.79 | 58.76 |
| | MSP+OE | 0.96 | 99.43 | 93.08 |
| | MSP+Ours | 0.73 | 99.66 | 96.83 |
| Textures | MSP | 48.10 | 87.98 | 59.17 |
| | MSP+OE | 2.44 | 99.05 | 93.46 |
| | MSP+Ours | 3.05 | 99.13 | 95.33 |
| SVHN | MSP | 20.96 | 92.73 | 65.29 |
| | MSP+OE | 1.60 | 99.20 | 92.58 |
| | MSP+Ours | 1.94 | 99.25 | 93.37 |
| CIFAR-100 | MSP | 48.56 | 87.29 | 55.53 |
| | MSP+OE | 21.65 | 94.89 | 81.57 |
| | MSP+Ours | 23.53 | 94.56 | 81.26 |
| LSUN-C | MSP | 15.22 | 95.42 | 78.18 |
| | MSP+OE | 3.51 | 99.15 | 95.15 |
| | MSP+Ours | 2.84 | 99.25 | 95.92 |
| LSUN-R | MSP | 27.31 | 91.43 | 64.92 |
| | MSP+OE | 2.74 | 99.20 | 94.35 |
| | MSP+Ours | 3.78 | 98.84 | 92.59 |
| iSUN | MSP | 27.36 | 91.48 | 64.65 |
| | MSP+OE | 2.71 | 99.17 | 94.30 |
| | MSP+Ours | 4.29 | 98.71 | 92.27 |
| Places365 | MSP | 50.82 | 87.40 | 57.33 |
| | MSP+OE | 10.10 | 97.60 | 90.82 |
| | MSP+Ours | 10.64 | 97.52 | 90.93 |
| Mean | MSP | 30.16 | 90.77 | 62.02 |
| | MSP+OE | 4.75 | 98.63 | 91.35 |
| | MSP+Ours | 5.21 | 98.62 | 93.03 |