# OpenReview forum: "Open-set Label Noise Can Improve Robustness Against Inherent Label Noise"
_NeurIPS.cc/2021/Conference — NeurIPS 2021 Poster_

### Official Review · Reviewer_5cBL · 2021-07-14

**Rating:** 6
**Confidence:** 4

**Summary:**

This paper proposed a training method for improving the robustness of deep neural networks against inherent label noise. Specifically, the authors first found that the open-set noise is helpful for the generalization and robustness of a deep neural network, and then proposed the strategy that use an additional training set with open-set noise as the regularization in the training objective. Experiments showed improved performance by the proposed method against inherent label noise.

**Limitations And Societal Impact:**

The authors did not mention the limitations of their method. However, in my opinion, the major limitation is the practical implementation issue as mentioned in the "Main Review" part, i.e., additional hyper-parameters and extra computational cost.

**Main Review:**

The counter-intuitive discovery that the open-set is not harmful but even helpful for the robustness of neural networks is interesting, and worth to be paid more attention to. Based on this, the proposed robust training method is simple to implement yet effective.

However, there still exist some issues to be addressed:

1. Theoretical justification. The authors provided theoretical analysis for the effectiveness of the proposed method from the perspective of SGD noise. However, it seems that this is not that essential, since there can be simpler method to add SGD noise other than introduce the additional data with open-set noise. The authors also explained the effectiveness by claiming that the capacity of the neural networks can be first consumed by the open-set noise and then the inherent noise. However, there is no theoretical guarantee for this.

2. Implementation issue. The proposed method introduces two additional hyper-parameters to be tuned, i.e, the size of the auxiliary dataset and $\eta$. While these hyper-parameters can be tuned by validation as mentioned by the authors, it could increase the computational cost. Besides, as can be observed from Figure 4 can claimed by the authors, the decreasing trend of test accuracy caused by inherent noise can be avoided with a larger $\eta$, which raises a question that what the performance could be if further increasing $\eta$. For the size of the auxiliary dataset, as can be observed from Figure 1 of the appendix, it affects the performance with a relatively large extent, while with no consistent trend, which increases the difficulty in tuning it. The authors also claimed that the proposed method is computational inexpensive, which I do not quite agree, since enlarging the training set can definitely increase the computational complexity.

3. Limited improvement on real world dataset. As can be seen from Table 3, the performance improvement solely by the proposed method is not that significant than solely by DivideMix. Considering the combination of the two methods, DivideMix improves the proposed method more than reverse.

**Time Spent Reviewing:**

8 hours.

---

> ### Author Response · Authors · 2021-08-10
> **Response to Reviewer 5cBL.**
>
> Thanks for your valuable comments. Please find our response below.
>
> 1. **Understanding from SGD noise is not important and there is no theoretical guarantee for our intuitive interpretation.**
>
> We strongly believe that the analysis from the perspective of SGD noise is important.
>
> * Our analysis builds a connection between open-set noises and SGD noises to help understand the effects of open-set noises during training. Without the analysis, the relations and differences between our work and existing literature would be ambiguous.
>
> * The reviewer raised an interesting point that there can be simpler methods to add SGD noise other than introducing open-set noises. However, it does not affect the importance of the analysis from SGD noise. As the SGD noises induced by our regularization are parameterized by model parameters and OOD data, they are fundamentally different from simple spherical Gaussian noises. Considering the dynamicity of our SGD noises, it would be very complicated to directly design SGD noises to simulate the effects of open-set noises.
>
> As for the intuitive interpretation, we explain the counterintuitive phenomenon in Section 2 by the “insufficient capacity” hypothesis, which has been thoroughly analyzed in existing literature [1]. This is also supported by our empirical analysis in Subsection 2.2.
>
> Additionally, there might be some misunderstandings. We would like to clarify that the claim from the reviewer “the capacity of the neural networks can be first consumed by the open-set noise and then the inherent noise” is inaccurate. In our statement, the order for the examples to be memorized is not necessary. Even if the open-set noises and the inherent noises could be memorized at the same time, increasing the number of open-set auxiliary samples would still slow down the fitting of inherent noises, as stated in the “insufficient capacity”.
>
> [1] Arpit, Devansh, et al. "A closer look at memorization in deep networks." International Conference on Machine Learning. PMLR, 2017.
>
>
> 2. **The hyperparameter optimization.**
>
> Thank you for pointing out the potential for misunderstanding. In our experiments, $\eta$ is the only additional hyperparameter to be tuned.
>
> For the size of the auxiliary dataset, we sample a batch of examples from the whole OOD dataset in each iteration by default. Figure 1 in the Appendix is to show that the sample size of the auxiliary dataset does not affect the performance too much due to Dynamic Noisy Labels. We can observe that using different-sized OOD datasets would achieve comparable improvements in performance, with a standard deviation of 0.41.
>
> As for $\eta$, we can observe from Figure 4 that with a large $\eta$ like 2.5, the decreasing trend of test accuracy caused by inherent noisy labels is nearly eliminated. However, It doesn’t simply mean a larger $\eta$ would be better. As shown in Figure 4, the performance slightly decreases when we increase the $\eta$ from 2.5 to 3.0. It is intuitive that if we set a value too large for the $\eta$, the network would put too much emphasis on fitting the open-set noises thereby downgrading the performance on the test set. Here, we also add more experimental results with larger $\eta$ to verify this point:
>
> | $\eta$ | 2.5 | 3.0 | 3.5 | 4.0 | 5.0 | 6.0 |
> |---|---|---|---|---|---|---|
> | Test Accuracy | 87.12 | 86.60 | 86.15 | 81.57 | 27.08 | 10.0 (Fail) |
>
>
> 3. **The proposed method is computationally expensive.**
>
> We agree that increasing the size of the training dataset would increase computational usage. However, our method does not introduce much additional computational cost compared to existing robust methods, which is explicitly shown in the table below. Here, we compare our method and existing methods on the average training times for each epoch. All the experiments are conducted on the CIFAR10 dataset with an NVIDIA GeForce RTX 3090.
>
> | Algorithms | Standard | Ours (All) | Ours (50k) | Decoupling | F-Correction | Coteaching | JoCoR | DivideMix |
> |:---:|:---:|:---:|:---:|:---:|:---:|:---:|:---:|:---:|
> | Training time (s) | 13.14 | 18.95 | 18.92 | 29.41 | 21.05 | 28.98 | 30.34 | 103.37 |
>
>
> From the table, we can observe that the training time of ODNL using the whole OOD dataset is comparable to that of ODNL using 50K examples as the auxiliary dataset. It is because we sample a batch of examples from the auxiliary dataset to join the training in each iteration, instead of simply combining the auxiliary dataset into the original training dataset. Thus increasing the size of the auxiliary dataset would not affect the computational usage. Overall, the proposed method is still a promising and computational inexpensive option for learning with noisy labels.
>
> 4. **Limited improvement with DivideMix on the real-world dataset.**
>
> In the experiments, we incorporate our ODNL method into existing robust training algorithms, showing that ODNL consistently improves existing methods and ODNL is complementary to existing robust algorithms. Here, we do not expect vanilla ODNL to achieve state-of-the-art results compared with many integrated methods, like DivideMix. These integrated methods are aggregations of multiple techniques while our work only focuses on one, therefore the comparison is not fair. Still, ODNL can be a promising option in the family of robust learning methods, because it can improve existing methods, as appreciated by the other reviewers.
>
> 5. **The authors did not mention the limitations of their method.**
>
> As mentioned in the checklist, the limitations and future directions of this work are presented in Appendix E, including instance-dependent open-set noisy labels and synthesized auxiliary datasets.

---

> > ### Comment · Reviewer_5cBL · 2021-08-17
> > **Still some concerns**
> >
> > Thanks for your clarifications, that have addressed several concerns of mine. However, there are still concerns remaining.
> >
> > Specifically, I do not think the parameter tuning issues have been well addressed.
> >
> > First, I do not think "Figure 1 in the Appendix is to show that the sample size of the auxiliary dataset does not affect the performance too much", since one percent in accuracy on CIFAR-10 dataset should not be neglected.
> >
> > Second, for the parameter $\eta$, the current manuscript only shows that a proper $\eta$ leads to a promising performance, but does not provide guidelines for tuning it, especially in real scenarios, where no clean data are available for validation. Besides, even if validation can apply, the time consumption could be not affordable, since one should run the whole algorithm for each value of $\eta$.

---

> > > ### Author Response · Authors · 2021-08-18
> > > **The concern on the parameter tuning.**
> > >
> > > Thanks for your prompt reply. Please find our response below.
> > >
> > > 1. **The effect of the sample size of the auxiliary dataset.**
> > >
> > > For the sample size of the auxiliary dataset, we use ***paired t-test*** at 0.05 significance level to formally check whether the differences among the performance of different sample sizes are statistically significant. In paired t-test, the experiments are repeated ten times with different seeds. We report all the test accuracies in Table 1 and the p-values for each pair of varying sample sizes in Table 2. From the table, we can observe that ***all the p-values of the pairs of different sample sizes are greater than 0.05***. That is to say, there is no evidence that would suggest that the sample size of the auxiliary dataset could affect the performance. Thus, we do not need to tune it in our experiments, which is consistent with our claim in Appendix D.3 (Line 118).
> > >
> > >
> > > 2. **The hyperparameter optimization.**
> > >
> > >
> > > Yes, we tune $\eta$ following a guideline that has been widely used in existing literature like SLN [1]. Given a new dataset with unknown noise, we suggest quickly searching the best $\eta$ (denoted as $\eta_o$) based on the binary search using the validation accuracy throughout training. 1) If we observe a decrease of validation accuracy at the late stage of training, it implies overfitting and $\eta < \eta_o$. 2) Otherwise, we have $\eta \geq \eta_o$. Based on 1) and 2), we can conduct a binary search to quickly find the best \eta even if one would like to search $\eta$ in a very detailed range.
> > >
> > > As for the validation set, the settings have been briefly introduced in Section 5. On CIFAR10 and CIFAR-100, following SLN [1] and GCE [2], we use 5K noisy samples (10% of the training data) to tune $\eta$. On Clothing1M, following SLN [1] and F-correction [3], we use the clean validation set to tune $\eta$.
> > >
> > > Finally, we would like to emphasize that tuning the hyperparameter for ODNL is easy compared with many baselines since we only need to tune $\eta$. And in our paper, we achieve good results by simply tuning $\eta$ in  {0.1, 0.5, 1, 2.5, 5}, as described in Subsection 5.1.
> > >
> > >
> > > ---
> > >
> > >
> > >
> > > Table 1: Test accuracy (%) on CIFAR-10 using ODNL $(η = 1)$ with different auxiliary dataset sizes under 10 different seeds.
> > >
> > > |Sample size|1|2|3|4|5|6|7|8|9|10|
> > > |:---:|:---:|:---:|:---:|:---:|:---:|:---:|:---:|:---:|:---:|:---:|
> > > |50K|82.28|82.05|82.12|82.35|81.88|83.43|82.06|81.38|82.19|83.31|
> > > |100K|83.22|83.38|83.04|82.80|83.52|82.51|82.30|83.06|80.95|82.66|
> > > |150K|82.65|82.44|82.55|82.25|83.22|82.54|82.99|82.82|83.08|82.72|
> > > |200K|82.72|81.98|82.44|82.18|82.37|82.79|82.48|83.42|83.05|82.29|
> > > |300K|82.24|83.85|81.53|82.12|82.69|82.44|83.12|82.32|82.37|82.63|
> > > |All|83.05|83.72|82.90|82.52|82.54|82.17|83.22|82.22|83.42|82.70|
> > >
> > > Table 2: P-values for pairs of different sample sizes in paired t-test.
> > >
> > > |Sample size|50K|100K|150K|200K|300K|All|
> > > |:---:|:---:|:---:|:---:|:---:|:---:|:---:|
> > > |50K|-|0.2247|0.1192|0.3434|0.4515|0.0850|
> > > |100K|-|-|0.9541|0.5957|0.4749|0.7485|
> > > |150K|-|-|-|0.2733|0.3671|0.5246|
> > > |200K|-|-|-|-|0.8819|0.2904|
> > > |300K|-|-|-|-|-|0.1132|
> > > |All|-|-|-|-|-|-|
> > >
> > >
> > > [1] Chen, Pengfei, et al. "Noise against noise: stochastic label noise helps combat inherent label noise." International Conference on Learning Representations (ICLR). 2020.
> > >
> > > [2] Zhang, Zhilu, and Mert R. Sabuncu. "Generalized cross entropy loss for training deep neural networks with noisy labels." 32nd Conference on Neural Information Processing Systems (NeurIPS). 2018.
> > >
> > > [3] Patrini, Giorgio, et al. "Making deep neural networks robust to label noise: A loss correction approach." Proceedings of the IEEE conference on computer vision and pattern recognition (CVPR). 2017.

---

> > > > ### Comment · Reviewer_5cBL · 2021-08-19
> > > > **Still issues**
> > > >
> > > > Thanks for your response. Now my concern on parameter $\eta$ no longer exists. However, for the size of auxiliary dataset, there are still issues.
> > > >
> > > > First, it seems that the average test accuracy reported in Figure 1 of the Appendix is not consistent with the data reported in Table 1 here. Specifically, I have calculated the average accuracy of Table 1, and obtained 82.31(50K), 82.74(100K), 82.73(150K), 82.57(200K), 82.53(300K) and 82.85(All), which is clearly different from that of Figure 1 of the Appendix. Therefore, it is not surprising that the statistical hypothesis test supports the authors' claim here. However, if using the original statistics that produced Figure 1 of the Appendix, the results might be different, since the average accuracy under each size varies more significantly in that figure.
> > > >
> > > > Second, t-test assumes samples are drawn from Gaussian distribution, while the obtained accuracy does not necessarily follow this assumption. Therefore, it is better to use non-parametric test, such as Wilcoxon rank-sum test.

---

> > > > > ### Author Response · Authors · 2021-08-19
> > > > > **The concern on the size of auxiliary dataset**
> > > > >
> > > > > Yes, the data reported for the paired t-test is not exactly the same as the average test accuracy reported in Figure 1 of the Appendix. This is because we repeat the experiments ***ten times*** with different seeds here to make the t-test more convincing statistically, while there are only five times in the experiments of Figure 1. According to the law of large numbers, the average of the results will tend to be closer to the expected value as more trials are performed.
> > > > >
> > > > > We will use the data reported here to update Figure 1 to avoid misunderstanding in the final version, although it does not affect the claim in Appendix D.3 (Line 118):
> > > > >
> > > > > > "In Fig. 1, we show that our method can use only 50,000 examples from 80 Million Tiny Images to achieve comparable performance with using all examples from this dataset",
> > > > >
> > > > > Thanks for your suggestion. We also conduct Wilcoxon rank-sum test on the data in Table 1 to check the significance of the differences. The results show *most* p-values of pairs of varying sample sizes *(13/15)* are greater than 0.05, which still support our claim.
> > > > >
> > > > > Finally, we need to emphasize that ***we do not tune the sample size of the auxiliary dataset and simply use the whole dataset in our experiments***. Therefore, the size of the auxiliary dataset is not an additional hyper-parameters to be tuned in our method and would not increase the computational cost. By the way, the discussion in Appendix D.3 is to give a guideline about how to choose or collect an open-set auxiliary dataset for applying our method in real applications, considering the sample size, realism, and diversity.

---

> > > > > > ### Comment · Reviewer_5cBL · 2021-08-19
> > > > > > **Concerns addressed**
> > > > > >
> > > > > > My concerns have been addressed.
> > > > > >
> > > > > > I suggest to more clarify the points you provide here about the auxiliary dataset for better guiding the practitioners.

---

> > > > > > > ### Author Response · Authors · 2021-08-20
> > > > > > > **Thanks for your recognition and suggestion.**
> > > > > > >
> > > > > > > Thanks for your recognition and suggestion. We are glad that you feel that the concerns about experiments have been well addressed. In the future version, we will extend the discussion about choices of auxiliary datasets in Appendix D.3, as you suggested.

---

### Official Review · Reviewer_z2XB · 2021-07-16

**Rating:** 6
**Confidence:** 4

**Summary:**

This paper empirically shows that open-set noisy labels can benefit the robustness against inherent noisy labels in the training data, and proposes a simple but effective regularization method through introducing open-set data with Dynamic Noisy Labels (ODNL) into the training process. Specifically, the ODNL utilizes the open-set data with randomly generated labels in training, to exploit the extra capacity of neural networks, and thus makes the network avoid the memorization of noises in training data. The model trained with ODNL can achieve a better capability on Out-of-Distribution detection tasks. Extensive experiments show the effectiveness of the proposed method.

**Limitations And Societal Impact:**

The authors have addressed some limitations about the work, like instance-dependent open noisy labels and using a synthesized auxiliary dataset. However, more discussions or possible solutions about the limitation of their method should be listed out, such as:
(1) The discussion of the ineffectiveness of using closed-set auxiliary data in training. Since the 80 Million Tiny Images dataset is collected from the web while CIFAR-5m is generated through Denoising Diffusion Probabilistic Models (DDPM), it is not sufficient enough to reach the conclusion.
(2) The intuitive interpretation of consuming ‘insufficient capacity' could be further explained by using models with different depths and widths.


**Main Review:**

This paper proposed a method for learning with noisy labels called Open-set data with Dynamic Noisy Labels (ODNL). Specifically, the ODNL utilizes the open-set data with randomly generated labels in training, to consume the extra capacity of neural networks, and thus make networks avoid the memorization of noises in training data. The model trained with ODNL can also achieve a better capability on Out-of-Distribution detection tasks. Extensive experiments show the effectiveness of the proposed method.

This work is somewhat novel since it first introduces open-set auxiliary datasets in learning with noisy labels. It also provides comparisons with other methods which deal with learning with noisy labels like label randomization and outlier exposure.
For the quality of this paper, the whole process is technically sound.

However, there are also some points that need to be improved.

First, the selection of a closed-set dataset should be reconsidered. Since the 80 Million Tiny Images dataset is collected from the web while CIFAR-5m is generated through Denoising Diffusion Probabilistic Models (DDPM), the comparison of using closed-set or open-set dataset may be not sufficient enough to reach the conclusion because we cannot make sure that images generated form DDPM could have an identical distribution as real data. Secondly, it would be better if the hyperparameter \eta is indicated in the experiment results. Thirdly, a curve of performance for models with different percentages of symmetric noise labels could better illustrate the improvement of the method. Finally, since an intuitive interpretation of this method is that it consumes insufficient capacity and makes the model hard to memorize the noises, a comparison experiment between using the proposed method on a small neural network, like ResNet-18, and a big neural network, like WRN, is quite meaningful. Here we can see that the improvement between standard method and standard method with ODNL in the paper (trained with WRN-40-2) is higher than those when trained with ResNet-34, which is shown in the appendix, under open-set and dependent noise, but lower under asymmetric noise.

And for clarity, this paper is well organized and written clearly. It is a bit of a pity that Table 3 is not close enough to its corresponding context, but that is okay.

As for the significance, the proposed method is simple while the improvement reported in the paper is quite impressive. This method could be a new way for learning with noisy labels if the claim of the ineffectiveness of using a close-set dataset as an auxiliary dataset could be further supported.


**Time Spent Reviewing:**

5

---

> ### Author Response · Authors · 2021-08-10
> **Response to Reviewer z2XB.**
>
> Thanks for your valuable comments. Please find our response below.
>
> 1. **The selection of the closed-set dataset for comparison.**
>
> Thank you for raising an interesting point here. We agree that the distribution of CIFAR-5m is not exactly identical to CIFAR-10, but they are very close for research purposes, as claimed in the paper [1]. For example, training on 50K examples from CIFAR-5m could achieve 91.2% test accuracy on CIFAR-10 with WideResNet28-10. Additionally, it is unrealistic to obtain an additional large-scale closed-set dataset with exactly identical distribution for our analyses or real applications, otherwise, we could directly apply semi-supervised learning algorithms to improve the performance. Therefore, CIFAR-5m is still an acceptable substitute for the close-set auxiliary dataset in our analysis.
>
> Despite this, we also conducted an additional experiment to further check the validity of the claim. Specifically, the CIFAR-10 dataset is split into two parts, where one part is used as the training dataset and the other part is used as the close-set auxiliary dataset. We also used 25K examples and 100K examples from 80M Tiny ImageNet as the open-set auxiliary datasets. Here, we conduct experiments in Subsection 2.2 with the above datasets and present the results in the table below. From the table, we can observe that the results with 25K open-set noisy labels are comparable to that of standard training, while close-set noisy labels severely damage the test performance. Thus the results still support our conclusion in subsection 2.2 and we will add the comparison into the final version.
>
> |  | Standard (Symmetric-40%) | + Close-set noises | +Open-set Noises (25K) | +Open-set Noises (100K)
> |:---:|:---:|:---:|:---:|:---:|
> | Test Accuracy | 61.79 | 45.21 | 61.47 | 64.18 |
>
>
> [1] Nakkiran, P., Neyshabur, B., & Sedghi, H. (2020). The Deep Bootstrap Framework: Good Online Learners are Good Offline Generalizers. ICLR, 2021.
>
>
> 2. **The value of the hyperparameter $\eta$.**
>
> The best value of the hyperparameter depends on the dataset, noise type (rate), network architecture, and the integrated method. For example, for training WRN-40-2 network on the CIFAR-10 dataset with symmetric-40% noises, we set $\eta = 2.5$ for vanilla ODNL and set $\eta = 2$ for ODNL + existing methods except JoCoR. For ODNL+JoCoR, we set $\eta = 0.5$. We will list the detailed value of $\eta$ for each experiment in the final version.
>
>
> 3. **A curve of performance for models with different percentages of symmetric noise labels.**
>
> We conduct experiments on CIFAR10 with different symmetric noise rates and show the test accuracies in the table below. We can observe that the proposed method consistently improves the test performance.
>
> | Noise rate | 0 | 0.1 | 0.2 | 0.3 | 0.4 | 0.5 | 0.6 | 0.7 |
> |:---:|:---:|:---:|:---:|:---:|:---:|:---:|:---:|:---:|
> | Standard | 94.96 | 89.88 | 86.05 | 82.76 | 77.55 | 70.01 | 61.32 | 45.42 |
> | Ours | 95.40 | 93.06 | 89.96 | 88.78 | 87.10 | 81.65 | 74.37 | 56.28 |
>
> 4. **Experiments with different network architectures.**
>
> Thank you for the suggestion. We conduct experiments on CIFAR10 (Symmetric-40%) with WRN-40-2, ResNet-18, and VGG-11, respectively. The results are presented in the table below. We can observe that the proposed method consistently improves the performance with the three different network architectures. Here, it might be hard to directly compare the improvements among the three network architectures, because Dynamic noisy labels in the proposed method would consume the extra capacity of the network no matter how large the network capacity is.
>
> Instead, we find an interesting point from the best choice of the $\eta$: The small networks like ResNet-18 and VGG-11 require smaller $\eta$ than the large network (WRN-40-2), as shown in the table. Generally speaking, the network would put more emphasis on fitting the open-set noises with a larger \eta. Therefore, the phenomenon shows that to reduce the overfitting of the inherent noisy labels, a large network needs to consume more capacity on the open-set noises than a small network, thereby supporting our interpretation.
>
> | Network | WRN-40-2 | ResNet-18 | VGG-11 |
> |:---:|:---:|:---:|:---:|
> | Standard | 77.55 | 62.62 | 62.04 |
> | Ours | 87.10 | 76.93 | 76.41 |
> | The best $\eta$ | 2.5 | 0.5 | 0.5 |

---

> > ### Comment · Reviewer_z2XB · 2021-08-20
> > **Post-rebuttal comment**
> >
> > Thank the authors for the detailed replies. My questions listed above are generally addressed.

---

### Official Review · Reviewer_wWfP · 2021-07-16

**Rating:** 5
**Confidence:** 4

**Summary:**

This paper provides an interesting perspective on how to deal with open-set label noise when learning with noise labels. In the noise label community, open-set noises are usually considered to be harmful for generalization, similar to closed-set noises. However, in this paper, the authors present a new perspective that the open-set noise can help generalization against inherent label noises by using proposed Dynamic Noisy Labels (ODNL) regularization on auxiliary open-set samples. In addition, using analysis from the lens of SGD noise, they empirically show the superiority of ODNL over other related regularizations such as Stochastic Label Noise (SLN) and Outlier Exposure (OE).

**Limitations And Societal Impact:**

An important related work that has a similar design and effect to the proposed method is missing.
There is no detailed description for Figure 2; what is x- and y-axis, how much auxiliary open-set noise data is used.
There might be a notation error in line 161. $f(\tilde{x}, \theta)$ seems to be $\tilde{f}(\tilde{x}, \theta)$.


**Main Review:**

The main contribution of this paper is that it is the first work to explore the benefits of open-set noise in learning with noisy labels. However, the authors have missed a very important paper related to their main argument.

[1] Lee, Saehyung, et al., "Removing Undesirable Feature Contributions Using Out-of-Distribution Data.", ICLR, 2021.

[1] argues that the open-set noise (i.e., out-of-distribution) can help to improve the generalization of DNNs by using uniform-label regularization on the auxiliary open-set noise samples. I think this uniform-label regularization has basically the same effect for improving the generalization of DNNs with ODNL, except its dynamicity.
Although [1] is not focusing on the label noise problem, I think the author should refer to the paper and clarify the difference and advantages against it. Also, the statement like “open-set noises are always considered to be harmful to the training of DNNs” should be modified.

Meanwhile, the analysis through SGD noise and comparisons to other regularizations such as SLN and OE are clear. In figure 3, with the convex combination of open-set noise and closed-set noise, the importance of open-set noise is clearly shown.
In the experiment, as shown in Table 1, the proposed method has advantages in its applicability to various existing methods because of its simplicity. Also, it showed a significant performance boost combined with other baselines. The out-of-distribution detection performance was also enhanced by their proposed method.

Although the authors provide insightful analysis and impactful methods that leveraging open-set noise to improve robustness against label noise, they missed out on an important related work that seems to have a very similar design and effect with the proposed one. In this concern, I recommend a score marginally below the acceptance threshold.


**Time Spent Reviewing:**

6

---

> ### Author Response · Authors · 2021-08-10
> **Response to Reviewer wWfP.**
>
> Thanks for your valuable comments. Please find our response below.
>
> 1. **Relations to a recent paper in adversarial training.**
>
> Thanks for bringing this recent paper in our attention. OAT [1] aims to use out-of-distribution (OOD) data to improve generalization in the context of adversarial robustness. While [1] is relevant to our work, we note that there are key differences in both technique and insight perspectives:
>
> * Technique: OAT regularizes the softmax probabilities to be a uniform distribution for out-of-distribution data, which is the same as Outlier Exposure (OE). As analyzed in Section 4 of our paper, the regularization in OE and OAT cannot improve the robustness against noisy labels, because it does not induce dynamic noises and the optimization of parameters always follows the direction of gradient descent.
>
> * Insight: OAT shows that OOD data samples share the same undesirable features as those of the in-distribution data so that these samples could also be used to remove the influence of undesirable features in adversarial robustness. In our work, we show that open-set noisy labels could be harmless and even benefit the robustness against inherent noisy labels. We give an intuitive interpretation for the phenomena by “insufficient capacity” and understand the effects of open-set noises from the perspective of SGD noises.
>
> We would like to clarify that our statement “open-set noises are always considered to be harmful to the training of DNNs” still holds, because we are the first to consider open-set noisy labels, where each OOD instance is assigned with a one-hot label. In contrast, OAT only uses uniform distribution as the targets of OOD instances (i.e., $t_{unif}=[\frac{1}{c}, \ldots, \frac{1}{c}]$, which is not one-hot label). Therefore, they cannot show "the open-set noise can help to improve the generalization"; instead, they can only show "out-of-distribution data with OE loss can help generalization". By the way, it has already been shown in the original OE paper that OOD data with OE loss is harmless to generalization.
>
> 2. **Detailed description for Figure 2.**
>
> In figure 2, the landscapes are visualized using the technique from the loss-landscapes library [2]. Specifically, the x- and y-axis represent the distances along with two random directions on the loss landscapes; the z-axis represents the loss value. By default, we sample a batch of samples from the whole OOD dataset in each iteration. We have included some descriptions for Figure 2 in Appendix C.2 and will add the above contents in the final version.
>
> 3. Thanks for pointing it out. We will fix these typos in the final version of the paper.
>
> &nbsp;
> &nbsp;
>
> [1] Lee, Saehyung, et al. "Removing Undesirable Feature Contributions Using Out-of-Distribution Data.", ICLR, 2021.
>
> [2] Li, Hao, et al. "Visualizing the loss landscape of neural nets." Neural Information Processing Systems, 2018.

---

### Official Review · Reviewer_aFDj · 2021-07-16

**Rating:** 7
**Confidence:** 3

**Summary:**

This work proposes augmenting a training set by using an out of distribution auxiliary dataset to help train a robust classifier through randomly sampling labels for the OOD dataset. Model's trained with this additional data perform better than related work on a variety of noisy classification tasks, including out of distribution detection.

**Limitations And Societal Impact:**

Weaknesses:
-The only weakness is the lack of novelty. This is a simple, yet effective, change to OE. And while a detailed analysis is performed, I would love to see the outlier dataset used more than just for crossentropy loss.

The authors do not address negative societal impact of their work. It could be mentioned how additional data increases compute/energy usage.

After seeing the author's response, and the rest of the discussion, I think the discovery that openset instance improve noisy label classification is definitely of interest to the community.  I've increased my score to reflect such.

**Main Review:**

Strengths:

-This proposed method is straightforward and effective

-A thorough analysis is done as to how ODNL effects models' parameters during training, specifically comparing to the most similar method Outlier Exposure.

-Many different experiments are used to demonstrate the effectiveness of ODNL.


Originality:
-The proposed method is fairly simple, that is, modifying Outlier Exposure to use random 1hot labels rather than the uniform distribution over all labels.

Correctness:
-Section 3 and 4 are well written and thoroughly analyze the effects of the proposed method.

Clarity:
-The paper is well written and easy to follow

Relation to prior work:
-This work appropriately places itself within the recent literature on noisy label classification.

Reproducibility:
-The source code is provided and the method seems easy to apply to any classifier. However, the used auxiliary dataset is no longer publicly available.



**Time Spent Reviewing:**

3

---

> ### Author Response · Authors · 2021-08-10
> **Response to Reviewer aFDj.**
>
> Thanks for your valuable comments. Please find our response below.
>
> 1. **The originality**
>
> While we appreciate the reviewer’s feedback, we respectfully disagree that the novelty of this work is simply a small change to OE. As presented in Section 4, OE is a method designed for OOD detection tasks and cannot improve the robustness against noisy labels. By contrast, our work is the first to find open-set noises could be harmless and the first to utilize an open-set auxiliary dataset in combating noisy labels. In addition, our originality also includes the analysis of the effects of open-set noises as a variant of SGD noises.
>
> Considering the simplicity and effectiveness of the proposed method, our work opens a new avenue to improve robustness against noisy labels, and this will inspire more specially designed methods using open-set auxiliary dataset to combat label noise in deep learning, in the future.
>
> 2. **Computational Usage**
>
> Thank you for this suggestion. In the table below, we compare our method and the existing robust methods on the average training times for each epoch. All the experiments are conducted on the CIFAR10 dataset with an NVIDIA GeForce RTX 3090. The results show that our method introduces negligible additional computational cost compared to the existing methods. We will include the comparison in the final version.
>
> | Algorithms | Standard | Ours(All) | Ours(50k) | Decoupling | F-Correction | Coteaching | JoCoR | DivideMix |
> |:---:|:---:|:---:|:---:|:---:|:---:|:---:|:---:|:---:|
> | Training time (s) | 13.14 | 18.95 | 18.92 | 29.41 | 21.05 | 28.98 | 30.34 | 103.37 |
>
> *Here, “All” and “50K” mean the size of the auxiliary dataset.*

---

### Author Response · Authors · 2021-08-25
**Looking forward to receiving feedback from Reviewer wWfP**

We thank all the reviewers for your recognition and encouragement. Thanks to the rebuttal, all concerns of three reviewers have been well addressed in discussion, while we are still looking forward to receiving feedback from Reviewer wWfP. Specifically, the main concern from Reviewer wWfP is about the differences between our work and OAT [1]. Here, we authors would like to emphasize that OAT is clearly different from our method since it uses the same way as OE [2] to utilize out-of-distribution instances for improving adversarial defenses. It is worthy to note that the differences between our method and OE have been clearly analyzed in our paper, as also appreciated by Reviewer wWfP. The detailed comparison can be found in our response to reviewer wWfP and Section 4 in the paper.

We sincerely thank you again for your very constructive reviews. Please kindly let us know if there are still unclear parts to you. We are very glad to further discuss them.

[1] Lee, Saehyung, et al., "Removing Undesirable Feature Contributions Using Out-of-Distribution Data.", ICLR, 2021.

[2] Hendrycks, Dan, Mantas Mazeika, and Thomas Dietterich. "Deep anomaly detection with outlier exposure." ICLR, 2019.

---

### Decision · Program_Chairs · 2021-09-27

**Decision:**

Accept (Poster)

**Comment:**

UPDATE: The revision from the authors has been reviewed.  After some back-and-forth with the authors to discuss the details of the 300K Random Images dataset that they have chosen to use, the paper has been officially accepted.

----

Reviewers generally appreciated the paper's observation regarding the benefit of open-set label noise as being interesting and non-obvious. The simplicity and effectiveness of the technique, coupled with the theoretical analysis, were also appreciated. Some concerns were however raised on:

(1) choice of hyper-parameters

(2) relation to OAT and other out-of-distribution regularization schemes

Point (1) was convincingly addressed in the response. There was some debate about point (2). From my reading, I do agree that there are conceptual similarities between the proposed technique and OAT. However, I also take the authors' point about the similarity being closer to OE, a baseline which the authors discuss and compare against. Further, even when considered as a variant of OAT for the label noise setting, the present paper contributes to theoretical understanding of such out-of-distribution regularization schemes.

The authors are encouraged to incorporate the reviewers' suggestions, include a discussion of OAT, and contrast their method to other regularization schemes for label noise (Wasserstein Adversarial Regularization (WAR) on label noise, ICLR 2020; Robust early-learning: hindering the memorization of noisy labels, ICLR 2021).

It was discovered late in the review process that this paper makes use of the 80 million tiny images dataset, which has been retracted (https://groups.csail.mit.edu/vision/TinyImages/).  Following the NeurIPS ethical guidelines (https://neurips.cc/public/EthicsGuidelines), this dataset should not be used.  For this reason, the paper is being conditionally accepted.